# Design Consideration of ZVS Single-Ended Parallel Resonant DC-DC Converter, Based on Application of Optimization Techniques

Nikolay Hinov [1,*] and Bogdan Gilev [2]

1   Department of Power Electronics, Technical University of Sofia, 1000 Sofia, Bulgaria
2   Department of Mathematical Modeling and Numerical Methods, Technical University of Sofia, 1000 Sofia, Bulgaria; b_gilev@tu-sofia.bg
*   Correspondence: hinov@tu-sofia.bg; Tel.: +359-2965-2569

**Abstract:** The paper presents the design of a single-ended transistor zero-voltage switch (ZVS) parallel resonant DC-DC converter. Due to the use of a resonant inverter in the structure of the DC-DC converter, it is characterized by high efficiency and improved performance. On the other hand, due to the specifics of the power circuit operation, in the work, it is proposed to find the values of some of the elements of the circuit of the electronic converter to be carried out based on the application of optimization. To solve this task, various tools available in Matlab/Simulink have been applied, as well as author's programs specially developed for the purpose. The use of a hybrid method for the design of power electronic converters, which combines analytical and optimization approaches, is justified in cases where there is no adequate design procedure. With the increase in the complexity of the power topologies and their possible modes of operation, difficulties arise related to their design such as: assumptions and limitations in conducting the analysis and the corresponding methodologies based on this analysis; high order of the differential equations composing the mathematical models; need for highly qualified specialists in the field of design. The proposed approach does not negate the classical design methods based only on analytical ratios determined by analysis of power circuits, but complements and develops them with innovative ones based on the application of computational mathematics and information and communication technologies.

**Keywords:** design consideration; single-ended resonant converters; ZVS; model-based design; optimization techniques

## 1. Introduction

Automotive electronics is an area of electronics specialized in the development, implementation and implementation of electronic systems and components used in the automotive industry. Automotive electronics include various aspects such as engine management, safety, comfort, entertainment and communications [1,2]. The main components and systems that are subject to automotive electronics are: electronic engine management; anti-lock braking system (ABS); Electronic Stabilization Program (ESP); navigation system; air conditioning system; audio and entertainment systems; electronic security systems. These are just some of the basic examples of automotive electronics applications [3,4]. Due to the continuous development of materials, elements, technologies and systems, an even greater variety of applications of automotive electronics is expected in the future, especially from the point of view of smart and autonomous vehicles. Characteristic of all these diverse in their structure, software and hardware provisioning systems is that they need power.

Power electronic devices in a vehicle are responsible for managing and controlling various electrical and electronic systems that require high levels of energy or power. These devices play an important role in the efficient operation of the car and include the following

components: alternator; rechargeable battery; starter. These are just some of the main applications of power electronic devices in the car. There are other systems and components such as electronic brakes, cruise control systems, traction control systems and others that use power electronics to provide better vehicle performance and safety [5,6].

Power electronic devices play a key role in electric vehicles (EVs) that use electric power for propulsion. These electronic converters are used to control the energy flows between the electric motors and the batteries in the EVs and include the following basic components [7,8]: An inverter that converts the direct current (DC) from the EVs battery to high-frequency alternating current (AC) to drive the electric motor. The inverter controls the frequency, voltage and phase of the alternating current to ensure smooth and efficient motor control; DC-DC converter, which is used to convert the voltage from one level of direct current to another level, which is suitable for charging secondary batteries or powering low-voltage electrical systems in EVs; a charger that converts the external power source (e.g., mains) into direct current with suitable parameters to charge the EVs batteries. This function is usually performed by a power electronic converter system that includes inverters and DC-DC converters to provide the required power level; a regenerative brake, which through the power electronic converters, transforms the kinetic energy when stopping or reducing the speed into electrical energy and then returns it to the EV's batteries. This allows the EVs to recover some of the energy that is lost in conventional braking using mechanical friction, thereby increasing the overall energy efficiency of the vehicle.

In this sense, the power electronic converters in the EVs are essential for the control of the electric motor, the batteries and the charging of the EVs. They enable more efficient use of electrical energy and provide greater autonomy and better performance for electric vehicles. Since power electronic devices play a key role in the efficiency, performance and convenience of EVs, more and more developments and innovations are being made in this area in order to improve the use of electrical energy and thus increase autonomy of EVs. In this way, the development and development of power electronic devices with application in vehicles is a current topic, the subject of numerous scientific studies and publications. On the other hand, due to the constantly increasing requirements and standards in the automotive industry, the indicators that must cover the power electronic devices used in vehicles are also increasing. This necessitates the search for power schemes that, in addition to being highly efficient, have improved characteristics and guaranteed performance. On this basis, the single-transistor schemes of resonant converters have become widespread. Initially, they were developed and implemented for the implementation of induction hobs and also for high-frequency heating of fluids [9–11]. It is characteristic of these converters that, under certain conditions, they work with direct establishment of the mode—that is, there are no transient processes during start-up. This is a very essential advantage that ensures the stability of the devices and greatly facilitates the synthesis of control. In this sense, due to their wide distribution and cost advantages, special designs of semiconductor elements have been developed to meet the specific needs and requirements of these devices [9,12]. In [13,14], various schematic options for the implementation of induction hobs are considered, basic design ratios are given and a method for recognizing the load is proposed, thanks to which the device operates with minimal losses. A natural development of single-ended transistor DC/AC converters are DC/DC converters. They are obtained on the basis of using an inverter transformer, a rectifier and a filter. Different systems for wireless charging of electric vehicles have been studied, with operating powers and frequencies being tailored to the characteristics of the element base [15–17].

A highlight of the presented manuscripts is the improvement of the performance of power electronic devices, compared to the AC mains supply, based on the application of active power filters (PFC). In [18–20], charge converters built on the basis of semiconductors with a wide band gap (SiC) are considered. The specifics of the construction and prototyping of the converters have been examined in detail, with the aim of achieving maximum efficiency, improving cooling and realizing reliable operation. In [21], a method for cooling a single-ended transistor converter for charging electric vehicles is proposed, which

improves cooling and reduces the cost of the device and operating costs. In [22], an optimal design of a wireless charger for an electric car was proposed, the main results being related to the application of new types of power transistors (SiC-MOSFET and Si-IGBT) and the provision of favorable temperature regimes. Single-ended transistor resonant converters in wireless power transmission systems are considered in [23,24]. An analysis and design of a parallel single-ended transistor circuit is proposed, and analytical dependences of the change in efficiency and output power when changing the load and the air gap are given. One development of the concept of applying single-ended transistor resonant converters is the creation of bidirectional converters for the implementation of the very topical Vehicle to Home (V2H) technology [25,26]. Experimental studies are presented that prove the low losses and the high efficiency of energy flow management realized with single-ended transistor resonance circuits. The following conclusions, recommendations and generalizations follow from the review, systematization and analysis of published manuscripts devoted to research related to single-ended transistor resonant converters [13,18,24–29]:

- Single-ended transistor resonant converters are realized according to two main topologies depending on connection diagrams of the resonant capacitor and the resonant inductance: series and parallel. Due to the peculiarities in the principle of operation and for the implementation of a mode with soft commutations by voltage, the series circuits are applied when working with low-resistance loads, and with parallel ones, it is possible to work with a wider range of changes in the load;
- Single-ended transistor circuits have a number of advantages compared to classical circuits of resonant converters (full-bridge, half-bridge with split resonant capacitor): power circuit with as few elements as possible; lack of transient processes during start-up (direct establishment of the mode); combining transistor control and protection; output power regulation based on transistor cut-off current; good dynamic performance and resistance to disturbances and load changes;
- The disadvantages of the considered topologies are related to the greater current and voltage load of the circuit elements, the asymmetric load of the DC power source and the generation of electromagnetic disturbances;
- It is expedient from an economic and operational point of view to design and prototype devices with a maximum power of 2–3 kW, based on IGBT transistors specially designed for the needs of these topologies;
- Several stages are observed during the period of operation of the power circuit, and the switching of these stages is not only a function of the external control implemented through the controller, but also depends on the operating mode. In this sense, the analyses and the design methodologies created on their basis are based on complex analytical relationships, which are difficult to apply both to the needs of education and engineering practice. On the other hand, the presence of tolerances of the resonant elements (usually in the range of 10–20%) renders the application of precise methods pointless, which are associated with heavy calculation procedures and high requirements for the hardware and its software.

On the other hand, due to the sinusoidal shape of the current in the AC circuit and of the output voltage, in the analysis and design of resonant converters, the method of the first harmonic and graphical dependences is often used [30–34]. The application of this approach is suitable for large values of the quality factor of the equivalent resonant circuit and when operating with control frequencies close to the resonant one. In [30–32], the design of converters that use higher-order resonant circuits is considered. Design ratios are derived that represent the AC inverter circuit as a second-order series resonant circuit. As a design aid, graphical dependencies are presented to determine the optimal working area. The main design task is to ensure soft switching operation of semiconductor devices when the load changes. Another characteristic feature of the presented analytical ratios is that when obtaining them, the losses in the resonant circuit are not taken into account, which are formed both by the resistances of the resonant elements and by bringing the load to the circuit. The use of increasingly complex topologies and the presence of

magnetic components in resonant converters makes the application of computer methods for analysis and simulation attractive. In [33], based on a model, simulation studies of the ZVS Parallel Quasi Resonant Converter were carried out, for the realization of an induction cooker. The idea of the authors is to avoid the use of complex procedures for determining the fundamental parameters necessary for the design of the device through numerical experiments with the model. Due to the specifics of the used power source—a photovoltaic generator—the research sought to work with maximum efficiency, and in this connection, graphical dependencies are given to support the design. In [34] a pragmatic approach for the automated design of Flyback Step-up DC-DC Converters is presented. Software is presented, which automatically determines the design parameters of the high-frequency transformer by searching a database of characteristics of the magnetic core provided by manufacturers. With another program, based on the calculated magnetic components, the losses in the circuit elements and the maximum and minimum current of the transistor are determined. Analytical approaches to the design of power electronic devices are usually extrapolated by assuming linear characteristics of the building elements and completely neglecting their tolerances. The main problem in the analysis and design of resonant converters is the strong dependence of their operating mode on the tolerances of circuit elements. In this aspect, the main problems in their construction and prototyping are related to the reliability and guarantee of their indicators [35,36]. Most often, these problems are solved by using an assessment of the residual resource of building elements depending on the conditions of their operation [35] or by statistical methods and probabilistic analysis [36]. A natural development of these methods is the tolerance analysis of power electronic devices combined with model-based optimization [37].

The motivation and purpose of this manuscript is to present a rational design procedure for a ZVS single-ended parallel resonant DC-DC converter based on the joint use of analytical dependencies, data accumulated from previous design experience, and mathematical optimization. A major tool for achieving this goal is the application of the modern toolset of modeling, computational mathematics and software engineering. The main goal of the research is to propose an optimization procedure that is applicable to all types of power electronic devices, regardless of their operating modes, used element base and circuit technical features. In this sense, an optimization based on the achievement of a set dynamics of the device is formulated and used. The often applied optimization, which seeks to achieve maximum efficiency of power electronic converters, is realized in a different way for each individual device and topology, since the losses in the circuit elements are determined with different analytical and graphical dependences.

The work has the following organization: the first part describes the basic applications, requirements and problems of automotive electronics. Special attention is given to the development of power electronic converters and systems and their contribution to improving the characteristics of electric vehicles; The second part examines the model-based approach to optimal design in power electronics; The third part describes the operation, the basic equations that are used to design and the mathematical model of a parallel resonant single-ended transistor DC-DC converter; In the fourth part, the first optimization task was formulated and solved on the basis of built-in functions of Matlab/Simulink, through which the values of the elements in the output circuit of the device were determined; In the fifth chapter, a second optimization problem based on solving multi-criteria optimization is formulated. To solve it, an author's program was developed and applied, which determines the optimal values of the same circuit components as in the previous section; The last sixth and seventh parts present a discussion, conclusions and directions for future research development in subsequent manuscripts.

## 2. Materials and Methods

Model-based design (MBD) of power electronic devices is a method that uses mathematical models and simulations for design, analysis and optimization. This approach offers

a number of advantages, including greater accuracy, shorter design time and lower costs. Here are some key elements of model-based power electronics devices design [38,39]:

Mathematical models—these are used to simulate and analyze the behavior of power electronic devices. These models describe the physical principles, electrical properties, and interactions between device components.

Simulation software—it is applied to perform simulations. Specialized software tools allow engineers to enter mathematical models and simulate the performance of power electronic devices under various conditions and loads.

Performance analysis—various types of performance analysis of power electronic devices are performed through model-based design. This includes evaluation of efficiency, thermal behavior, power management and other parameters essential to device performance, operation and reliability.

Optimization and design of the device—this is the final stage of the process, which is carried out after the computer simulations and analyses. In this sense, MBD enables engineers to optimize the design of power electronic devices, applying various optimization procedures and objective functions. Designers can experiment with different configurations, parameters, and control strategies to achieve desired goals, such as increasing efficiency, reducing energy losses, improving performance, and more.

Model-based design of power electronic devices is a powerful tool that helps engineers create better and more efficient devices. This method reduces the need for physical prototypes and trials, which saves time and resources in the development, prototyping and commercialization process and enables faster implementation of new technologies and improvements in power electronics.

Model-based optimization of power electronic devices refers to the use of mathematical models and algorithms to find the best device parameters and configurations. This approach allows engineers to find optimal solutions that meet set goals and constraints. Here are some key aspects of model-based optimization in the context of its application in power electronic device design:

Definition of objective functions: First, objective functions must be defined that relate to the performance, efficiency, reliability, or other important parameters of the power electronic device. This can include maximizing efficiency, reducing energy losses, minimizing heat load, and more.

Mathematical models and simulations: To conduct the optimization, mathematical models of the power electronic device are used, which describe the relationships between the parameters and the objective functions. These models are used to perform simulations where the parameters are varied and their impact on the objective functions is analysed.

Optimization algorithms: Various optimization algorithms are used to find the optimal parameter values. These algorithms can be genetic algorithms, particle swarm algorithms, gradient descent methods, and others. The goal is to search the space of possible parameter values and find those that maximize or minimize the objective functions.

Constraints and conditions: Constraints and conditions that may limit the allowable parameter values should also be considered in the optimization process. This may include limits on voltages, currents, heat loads, weight and dimensions, and more.

Iterative process: Model-based optimization is typically an iterative process where successive iterations of simulations and optimization algorithms are performed. Each iteration seeks to find better parameter values and improve the objective functions until the desired optimal solution is reached.

The work proposes a rational design approach based on the following combination of techniques: key ratio analyzes to find the values of the inverter circuit elements; setting a range of values and modeling and optimization to obtain the final values of the filter components in the converter output.

The advantages of the model-based approach for the optimal design of power electronic devices will be demonstrated by investigating a ZVS parallel resonant single-ended transistor DC-DC converter. Figure 1 shows the power diagram of the device. It includes

two types of converters: a resonant inverter composed of a transistor *T*, a reverse diode *D*, a resonant inductance *L*, a resonant capacitor *C* and a rectifier with a CLC filter (filter capacitances $C_2$ and $C_1$; filter inductance $L_3$). A high-frequency transformer is used to match the inverter output and the load requirements at the output of the device *R*.

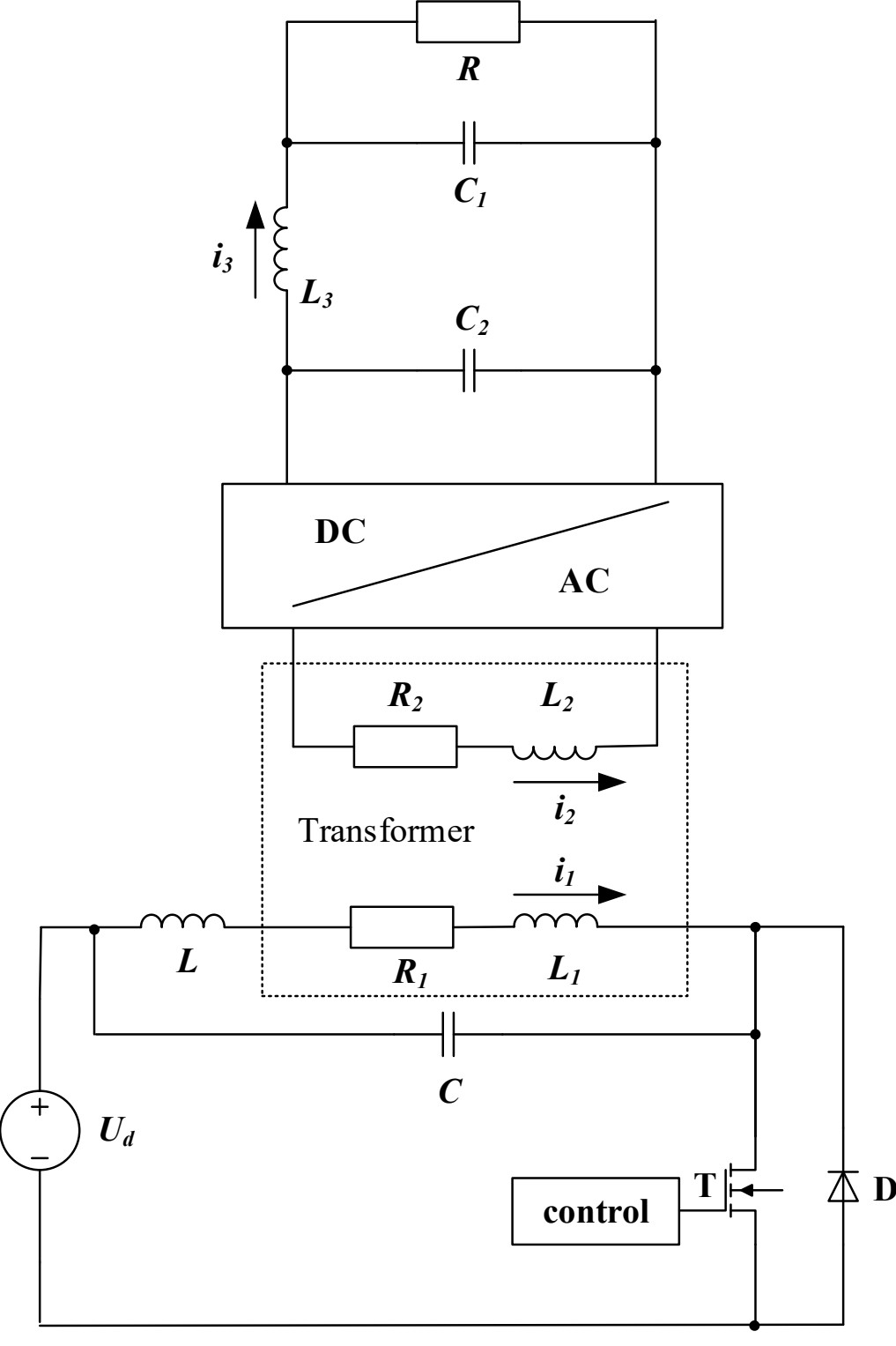

**Figure 1.** ZVS Single-ended transistor resonant DC-DC converter with parallel resonant circuit.

The analysis and the device model created on this basis were carried out while neglecting the losses of all building elements of the power circuit.

The choice to apply model-based to this topology was made because no design methodologies have been developed for it due to the complexity of the electromagnetic processes on the one hand, as well as the high order of the differential equations that define the state variables. Applying a hybrid approach to design is justified because the creation of a methodology requires a lot of effort both for its development, implementation and adaptation and for power electronics training purposes.

### 3. Description, Basic Relationships and Modeling of a Converter

The power circuit of Figure 1 is a combination of a parallel single-ended transistor resonant DC/AC converter with direct transition to an established mode of operation (no start-up transient), a transformer connected through its primary side to the resonant circuit of the inverter, and its secondary winding is connected AC/DC converter, CLC output filter and load. Single-ended transistor resonant inverters are usually designed to operate in (ZVS) mode, which improves their energy performance. On the other hand, due to the presence of resonant processes in a parallel circuit, significantly lower values of the voltage on the semiconductor switch $T$ are obtained (in the range of 2–3 times the input supply voltage $U_d$) compared to the series resonant inverter considered in [29,40]. This is also one of the main reasons for the wider application of this type of topologies compared to sequential ones. To control this type of power electronic converters, a method of monitoring the current of the power transistor is used [29], and its shutdown is carried out at a certain preset value of the current. In this way, control and protection of the converter are combined. On the other hand, the considered topology has very good technical, economic and operational indicators and is therefore applied in many places where a low-budget, efficient and reliable solution is sought.

Since the considered power devices have a complex structure with several stages during the period of their operation, it is difficult to analytically present the electromagnetic processes in the power circuit, and it is very difficult to create and apply basic design relations.

Essentially, the object of study is a power electronic system composed of two separate converters: a parallel resonant DC-AC converter and an AC-DC converter with output filter and load. Due to the complexity of the studied structure, a combined approach will be used in the design—the elements of the inverter will be determined by analytical ratios, and the elements of the output filter by applying model-based optimization in the Matlab environment.

An analysis of the parallel resonant DC-AC converter is discussed in [10,13,41,42]. Two stages of the inverter operation are usually distinguished: when the transistor/reverse diode structure works; and a resonant stage in which neither the transistor nor the reverse diode in the inverter conduct. Figure 2 shows diagrams explaining the operation of the parallel resonant inverter. The condition for direct establishment of the mode is that, at the initial switching on of the power circuit, the parallel resonant capacitor $C$ is charged with a voltage equal to the value of the DC power source $U_d$ [10,27]. The starting moment is the one when the transistor $T$ is turned off (moment 0 from Figure 2). Then, due to the stored energy in the resonant inductance, a resonant process begins in the parallel RLC circuit. This resonant circuit includes the following elements: the equivalent resonant inductance (the inductance $L$ from the inverter plus the total inductance of the transformer, rectifier and filter brought to the primary side of the transformer); the resonant capacitor $C$ and the active resistance of the device brought to the primary side of the inverter transformer. Due to the presence of resonance processes, the current through the resonant inductance and the voltage of the capacitor have a form close to sinusoidal.

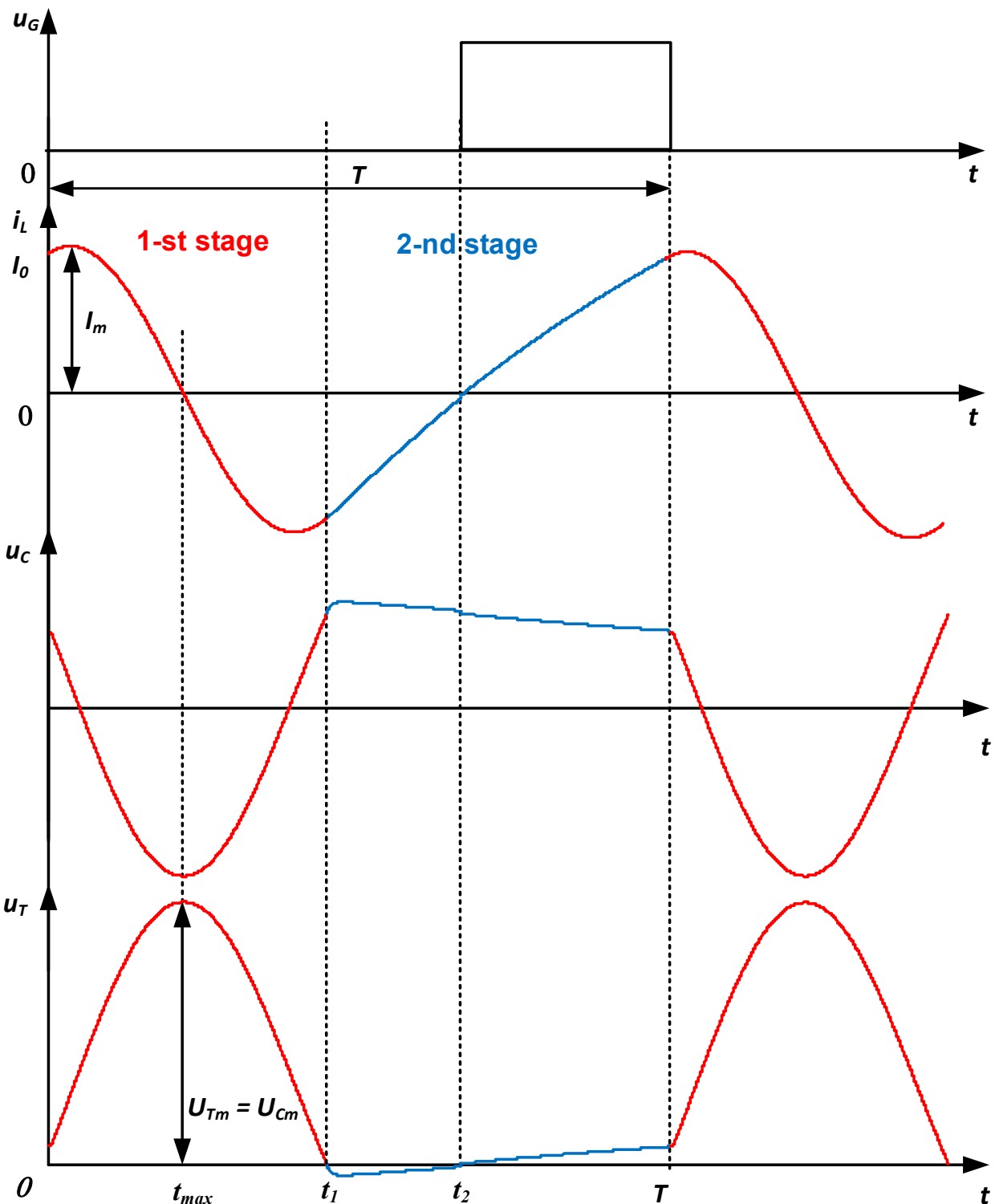

**Figure 2.** Timing diagrams describing the operation of a ZVS parallel single-ended transistor resonant DC/AC converter. From top to bottom, as follows: control pulses to the transistor, current through the resonant inductance, voltage across the resonant capacitor, and voltage across the transistor (and its reverse diode).

The determination of these two quantities is related to conducting an analysis of resonant inverters with a capacitor loaded in parallel, which is completed in detail in [9,27,40,42]. According to the description of electromagnetic processes in a parallel resonant circuit

with losses, for the current through the resonant inductance and the capacitor voltage is obtained as follows:

$$i_{LR}(t) = Ae^{-\delta t}sin(\omega_0 t + \psi)$$

$$u_c(t) = Ae^{-\delta t}\left[\frac{R_R}{2}(sin\omega_0 t + \psi) + \omega_0 L_R(cos\omega_0 t + \psi)\right] \tag{1}$$

where $L_R = L + L_1$—total inductance in the resonant loop, $R_R$—the total resistance in the resonant loop, $\omega_0 = \sqrt{\frac{1}{L_R C} - \delta^2}$—the resonant frequency and $\delta = \frac{R_R}{2L_R}$—attenuation of the parallel resonant circuit.

The constant $A$ and the initial phase of the load current $\psi$ are determined by the initial conditions of the resonant current $I_0$ and the capacitor voltage $U_0 = U_d$ and are:

$$A = \frac{2U_d - R_R I_0}{2\omega_0 L_R}\sqrt{1 + tg^2\psi};\ I_0 = \frac{U_d}{R_R}\left(1 - e^{-\delta t_i}\right);\ \psi = arctg\frac{2\omega_0 L_R I_0}{2U_d - R_R I_0} \tag{2}$$

When substituting the initial values of the resonant current $I_0$ and the voltage of the capacitor $U_0$ determined in this way, Formula (1) takes the form:

$$i_{LR}(t) = \frac{2U_d - R_R I_0}{2\omega_0 L_R}\sqrt{1 + tg^2\psi}\,e^{-\delta t}sin(\omega_0 t + \psi) = \frac{U_d}{\omega_0 L_R}Be^{-\delta t}sin(\omega_0 t + \psi)$$

$$u_c(t) = U_d Be^{-\delta t}\left[\frac{\delta}{\omega_0}(sin\omega_0 t + \psi) + (cos\omega_0 t + \psi)\right], \tag{3}$$

where $B = \left(1 - \frac{A}{2}\right)\sqrt{1 + tg^2\psi}$.

The equation expressing the voltage of the resonant capacitor and the primary winding of the converter transformer is presented in the following condensed form, with the corresponding starting phase:

$$u_C(t) = U_d Ce^{-\delta t}sin(\omega_0 t + \psi + \phi) = U_d Ce^{-\delta t}sin(\omega_0 t + \varphi) \tag{4}$$

where $C = \sqrt{1 + \left(\frac{\delta}{\omega_0}\right)^2}B$, $\phi = arctg\frac{\omega_0}{\delta}$, $\varphi = \psi + \phi$.

The first stage of operation of the resonant inverter ends at time $t_1$. An indication of the occurrence of this new state of the power circuit is the inclusion of the diode $D$. With the assumptions made in this way to work with ideal elements, the moment of switching among the two states is determined by the condition of equality between the voltages of the resonant capacitor and the DC power source $U_d$. Thus, from the condition $u_C(t_1) = U_d$, the end of the first and the beginning of the second stage of the operation of the inverter is determined. This is how the equation is obtained, the solution of which is the sought quantity—the duration of the resonance process—$t_1$:

$$u_C(t_1) = U_d Ce^{-\delta t_1}(sin\omega_0 t_1 + \varphi) = U_d \tag{5}$$

After applying mathematical transformations, the duration of the resonant process in the inverter is determined by the following equation:

$$e^{-\delta t_1}(sin\omega_0 t_1 + \varphi) = \frac{1}{C} \tag{6}$$

During the second stage of work of the device, the current through the resonant inductance decreases to zero, thereby ensuring the zero turn-on current of the transistor.

During the conduction of the reverse diode, the inverter control system supplies control to the transistor, and when the load current is reset, conditions are created for the transistor to turn on, and the transistor begins to conduct. The value of the resonant

inductance current $I_0$ at the end of the second stage (when the transistor is turned off) is determined by the expression [27]:

$$I_0 = \frac{U_d}{R}\left(1 - e^{-\frac{\delta\pi}{\omega_0}\frac{4\gamma}{v}}\right) = \frac{U_d}{R}\left(1 - \left(\frac{k-1}{k}\right)^{\frac{4\gamma}{v}}\right) = \frac{U_d}{R}F \tag{7}$$

where $\gamma$ is the fill factor, representing the ratio of the turn-on time of the semiconductor switch ton, to the period of the switching frequency $T$, $k = \frac{1}{1-e^{-\frac{\delta}{\pi}\omega_0}}$—coefficient of oscillation, analogue of the quality factor of the resonant circuit $Q$, $v = \frac{\omega}{\omega_0}$ frequency factor (detuning of the resonant circuit), $\omega = 2\pi f$—circular control (switching) frequency.

From the laws of electrical engineering, it is established that at the moment $t_{max}$ when the resonant current becomes zero, then the voltage on the resonant capacitor reaches its maximum value. With the directions of the currents and voltages chosen in this way, it is clear from the time diagrams in Figure 2 that this is a negative value compared to the initial one. Thus, by Equation (3), for the resonant current in this interval, $t_{max}$ is found:

$$t_{max} = -\frac{\psi}{\omega_0} = -\frac{1}{\omega_0}arctg\frac{2\omega_0 L_R I_0}{2U_d - R_R I_0} = -\frac{1}{\omega_0}arctg\frac{\frac{\omega_0}{\delta}A}{2-A} \tag{8}$$

Therefore, for the maximum voltage value of the resonant capacitor $C$, we obtain:

$$U_{C_{MAX}} = u_C(t_{max}) = U_d C e^{-\delta t_{max}}sin(\omega_0 t_{max} + \varphi) \tag{9}$$

The voltage across the transistor and its reverse diode is found as the difference between the voltage across the DC power source and across the resonant capacitor $C$:

$$u_T(t) = U_d - u_C(t) = U_d - U_d C e^{-\delta t}sin(\omega_0 t + \varphi) \tag{10}$$

The maximum value of the voltage of the semiconductor elements will occur at the moment tmax, when the capacitor $C$ is also at maximum voltage:

$$u_{TMAX}(t) = U_d - u_C(t_{max}) = U_d - U_d C e^{-\delta tmax}sin(\omega_0 t_{max} + \varphi) \tag{11}$$

In this way, with the help of the presented expressions, all the quantities necessary for the design of the power converter are determined.

The mathematical model of the considered power scheme is described by the following hybrid system of differential and logic equations:

$$\begin{pmatrix} L_1 + L & L_m \\ L_m & L_2 \end{pmatrix}\begin{pmatrix} \frac{di_1}{dt} \\ \frac{di_2}{dt} \end{pmatrix} + \begin{pmatrix} R_1 & R_m \\ R_m & R_2 \end{pmatrix}\begin{pmatrix} i_1 \\ i_2 \end{pmatrix} = \begin{pmatrix} -u_C \\ u_{C_2}\,sign(i_2) \end{pmatrix}$$

$$L_3\frac{di_3}{dt} + u_{C_1} - u_{C_2} = 0$$

$$-\left|i_2\right| - i_3 = C_2\frac{du_{C_2}}{dt}$$

$$i_3 = \frac{u_{C_1}}{R_3} + C_1\frac{du_{C_1}}{dt}$$

$$C\frac{du_C}{dt} = i_1 - control(t)\frac{U_d + u_C}{0.1} \tag{12}$$

$$where\ control(t) = \begin{cases} 1,\ for\ \left|\begin{array}{l} 0 \le i_1 \le I_{off} \\ i_1 - increase \end{array}\right.\ or\ u_C + U_d \le 0 \\ 0,\ for\ all\ other\ cases \end{cases}$$

$$sign(i_2) = \begin{cases} 1,\ for\ i_2 \ge 0 \\ -1,\ for\ i_2 < 0 \end{cases}$$

where $i_1$ is the current in the primary side of the transformer, $I_{off} = I_0$—the braking current of the transistor; $u_C$—the voltage of the resonant capacitor; $i_2$ is the current in the secondary

winding of the transformer; $i_3$ is the current through the filter inductance $L_3$ at the output of the AC/DC converter; $u_{C1}$—is the voltage across the filter capacitor at the output of the AC/DC converter; $u_{C2}$—is the voltage across the filter capacitor in the output of the device.

The presented model was developed after applying Kirchhov's laws and the switching functions *control(t)* and *sign(i₂)*, through which the control and rectification of the current in the secondary side of the transformer are reflected, respectively.

The Simulink/MATLAB implementation of the model (12) is shown in Figure 3. In the particular case, a symbolic implementation of the model was chosen, which allows the use of the standard optimization functions built into the mathematical software used. This approach is also useful for the application of optimization procedures by non-specialists in applied mathematics, such as power electronics designers and also power electronics students.

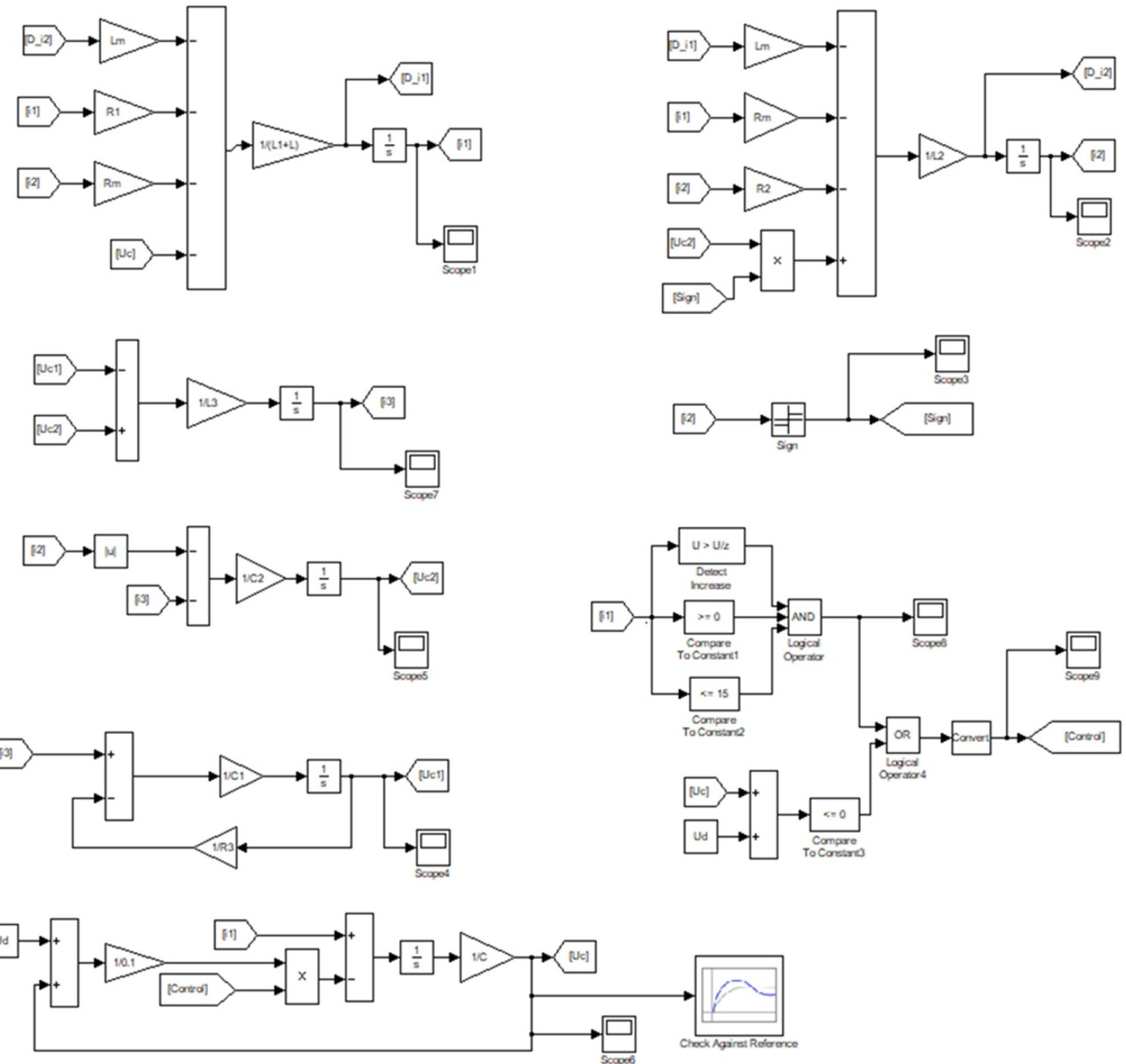

**Figure 3.** Realization of the model of ZVS single-ended transistor parallel resonant DC/DC converter in Matlab/Simulink environment.

The hybrid design approach of the investigated power electronic device will be applied in the development of a charger for an electric vehicle. Due to the specifics of the power

scheme, a maximum power of 2 kW and a control frequency change range of 20–30 kHz, an RMS value of the output voltage of 400 V was selected. In the example, power from the mains will be used, and to ensure good performance against the mains, a passive way to improve the power factor is used [10,17]. Another significant advantage of passive power factor correction, in addition to simple implementation and low cost, is the elimination of the need for a DC/AC converter start-up procedure, which is necessary to ensure soft voltage commutations and direct mode establishment. As a result of applying a design methodology, the elements of the inverter have been determined, and for the remaining circuit elements, based on experience gained from design and limitations related to the practical implementation of the devices, initial values have been indicated. The idea of the method is to determine these initial values based on the application of various optimization techniques. In this sense, the following values of the circuit components of the device are set:

$U_d$ = 205 V—voltage of the DC power source;

The inverter transformer parameters are as follows: $R_1$ = 0.01 Ω—resistance of the primary side;

$L_1$ = 0.425 µH—inductance of the primary side; $R_2$ = 0.01 Ω—resistance of the secondary side;

$L_2$ = 0.425 µH—inductance of the secondary side;

$L_m$ = 0.4225 µH—mutual inductance; $R_m$ = 0.0099 Ω—mutual resistance;

$L$ = 0.1 µH—additional inductance in the AC circuit of the inverter;

$C$ = 250 µF—resonant capacitor;

$R_3$ = 35 Ω—load resistance;

$L_3$ = 200 µH—filter inductance;

$C_1$ = 300 µF—filter capacity;

$C_2$ = 100 µF—filter capacity of the rectifier output.

In order to obtain a comparison of the results obtained through different optimization techniques, two optimization tasks were formulated and solved: The first was implemented in the Simulink/MATLAB environment, and the second was solved with an author's program, which describes the model presented in Figure 3.

## 4. First Optimization Problem

Optimization is a purposeful activity to obtain the best result in a certain sense and under set constraints. In this sense, optimal design refers to the process of creating a product, system, building or process that is optimal with respect to set criteria. The end result of optimization is expected to find the best solution that meets specific requirements and goals. The optimal design of power electronic devices is defined as an overall rational approach to create efficient, reliable, at the lowest possible cost and functional products. A very good review focusing on an application of optimization in power electronics based on various concrete examples is presented in [43].

Usually, very good results of applying optimization procedures to determine the values of the elements of the power circuits are obtained by setting a reference curve of the output voltage [37,44]. In this way, a certain dynamic of the device is set, which must be achieved as a result of the design. The degree of correspondence between the task and the achieved final result is evaluated with the *J(x)* criterion. This criterion is a number depending on the selection of the vector of parameters $x = [C_1, C_2, L_3]$, which represent the sought values of the circuit components of the CLC output filter.

The following optimization task was chosen—limiting voltage $u_{C1}$ in transient mode and preventing large pulsations of this voltage in established mode by optimal selection of filter capacitors $C_1$ and $C_2$ and inductance $L_3$. To realize this task, a suitable reference curve $u_{C1\_ref}$ is selected, whose analytical expression is:

$$u_{C_1, ref} = 400(1 - e^{-t/T}), \; for \; t \in [0, 0.02] \; and \; T = 0.003 \tag{13}$$

The selected trajectory will be used to search for a suitable value of the elements $C_1$, $C_2$ and $L_3$ so that the difference between the reference shape $u_{C1\_ref}$ and the shape of the output voltage $u_{C1}$, obtained by numerical experiments is minimal, i.e., to minimize the following functional:

$$J_1 = \int_0^{t_{end}} \left( u_{C1} - u_{C_1,ref} \right)^2 dt \underset{(C_1,C_2,L_3)}{\rightarrow} \min \tag{14}$$

This optimization assignment is solved under the following constraints: six equality constraints (12) and six inequality constraints (15).

$$\begin{matrix} C_{1,min} - C_1 \le 0 \\ C_1 - C_{1,max} \le 0 \end{matrix} \text{ and } \begin{matrix} C_{2,min} - C_2 \le 0 \\ C_2 - C_{2,max} \le 0 \end{matrix} \text{ and } \begin{matrix} L_{min} - L_3 \le 0 \\ L_3 - L_{max} \le 0 \end{matrix} \tag{15}$$

Equalities (12) are the equations describing the model, and inequalities (15) set the range of values of the parameters $C_1$, $C_2$ and $L_3$.

We obtained an optimization task that can be solved with a built-in Simulink/MATLAB procedure. For this purpose, the "Check Against Reference" block is added to the Simulink model (12), as shown in Figure 3. The specific setting of the optimization parameters is shown in Figure 4 (the parameters of the reference curve) and Figure 5 (the initial values and the limits of parameter variation).

The built-in Matlab/Simulink optimization procedure uses the gradient descent method [43,45]. The gradient descent method is an optimization algorithm used to find local minima or maxima of differentiable functions. It is widely used in machine learning and other fields of science where it is necessary to apply numerical methods. The main idea behind the gradient descent method is to find the direction of fastest decreasing gradient of the function under study. The optimization is realized by following this direction (of fastest decreasing gradient) at each step to update the current value of the variables. The gradient of a function at a given point indicates the direction of the fastest increase in the function at that point. Therefore, the opposite direction of the gradient points to the direction of the fastest decreasing function. The main steps to implement the gradient descent method are as follows:

1.  Initialize seed values for the variables you want to optimize.
2.  Calculating the gradient of the function at the current point. The gradient is a vector that contains the partial derivatives of the function with respect to all variables.
3.  Update the values of the variables by moving them in the opposite direction of the gradient. This is executed by multiplying the gradient by a fixed step, called the learning rate, and then subtracting from the current values of the variables;
4.  Repeat steps 2 and 3 until you reach a condition to stop the procedure, for example reaching a predetermined threshold number of iterations or if the gradient becomes close enough to zero.

An important aspect when using the gradient descent method is choosing an appropriate value for the learning rate. If the learning rate is too large, the algorithm may skip minima and not encounter convergence. If the learning rate is too small, the algorithm may move too slowly and take a long time to converge to the minimum.

Additionally, it should be noted that the main disadvantage of the gradient descent method is that it can be affected by the problem of local minima. This means that the algorithm can stay in a local minimum and not reach the global minimum of the function. The most common way to deal with this problem is by using a random starting point of the optimized quantities or combining it with other optimization methods.

Of course, the parameters that will be optimized correspond to the values of the elements of the CLC filter of the DC/DC converter, which were defined by inequalities (15).

After running the optimization, the results shown in Figure 6 are obtained. The determined optimal values of the output filter elements are also displayed on the screen.

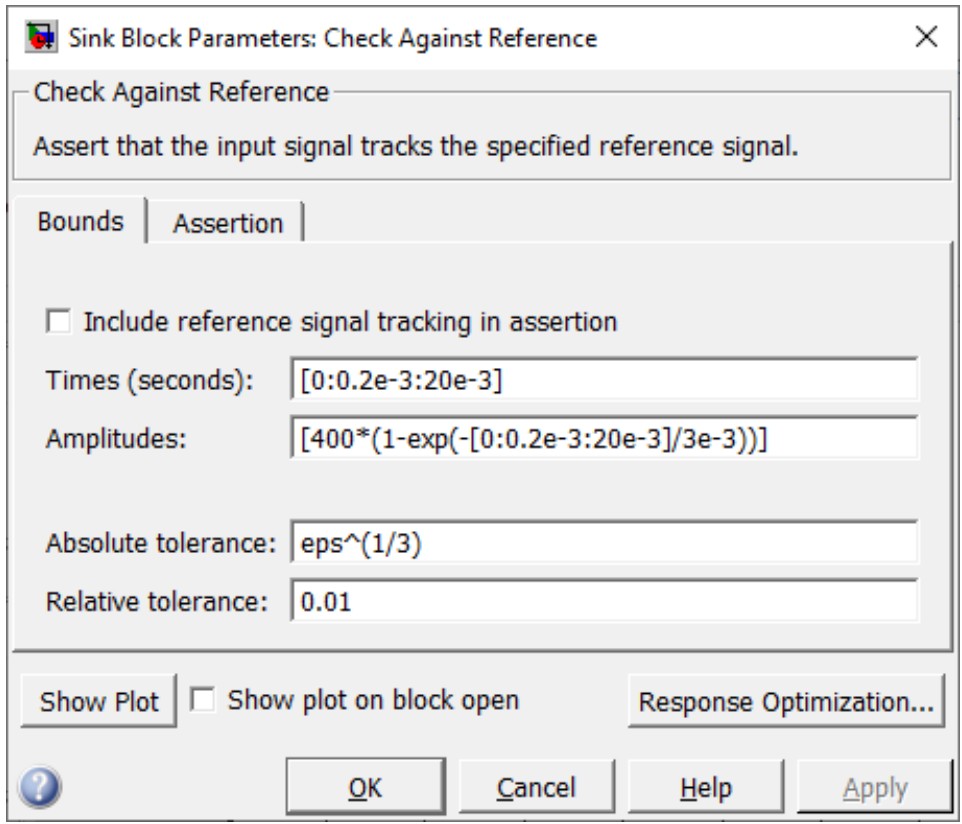

**Figure 4.** Set the reference curve used for optimization.

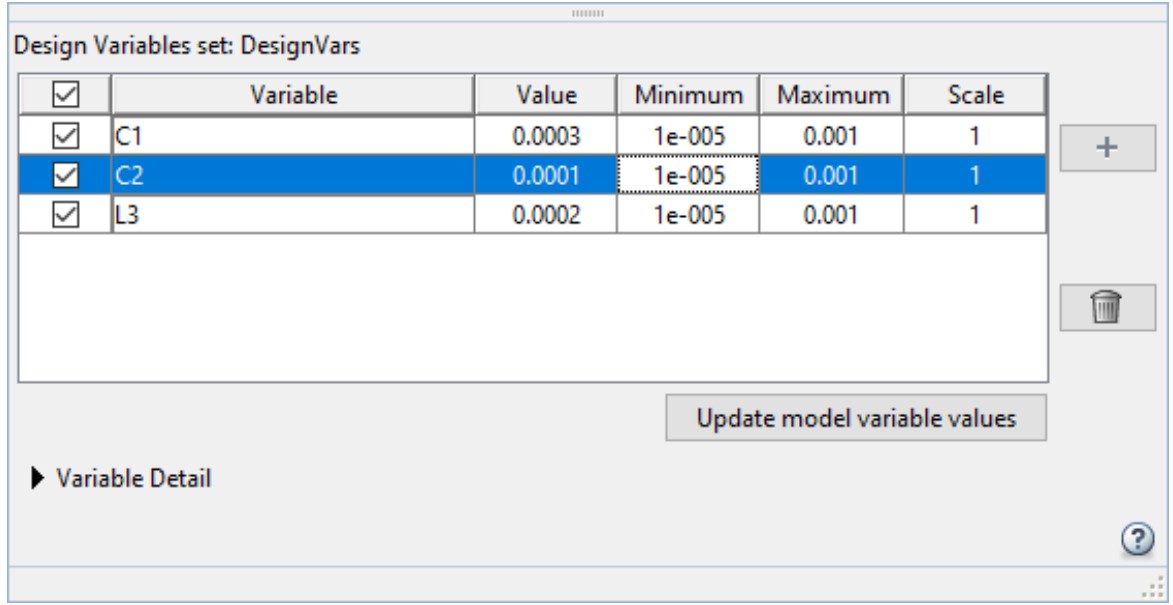

**Figure 5.** Setting the initial values and limits of variation of the parameters to be optimized.

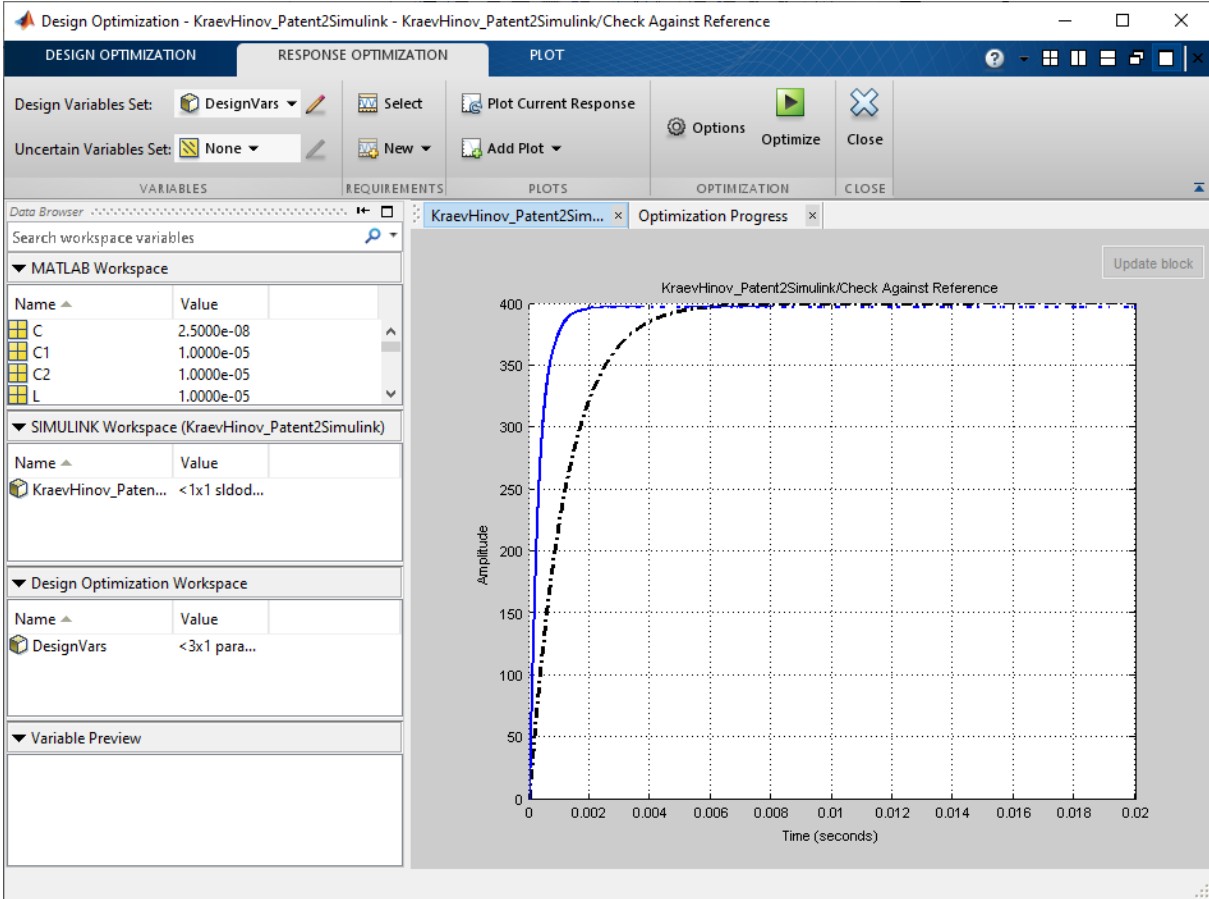

**Figure 6.** Realized and reference trajectories of the voltage $u_C$.

The following optimal values of the parameters $C_1 = 10$ μF, $C_2 = 10$ μF and $L_3 = 10$ μH are obtained. With the obtained optimal values, the output voltage and current are simulated and the results of Figure 7 are obtained.

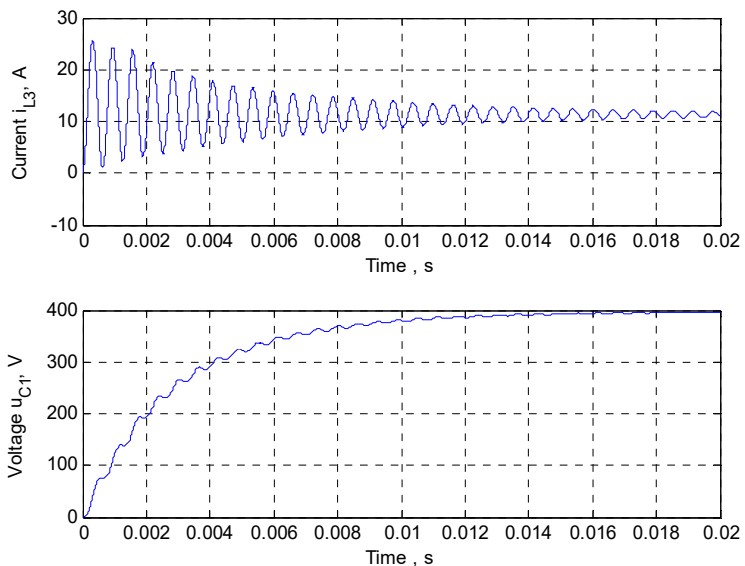

**Figure 7.** Results of simulation studies of an optimized power electronics converter: **top**—current through the output inductance and **bottom**—voltage at the output of the device.

From Figure 7, it can be seen that the desired goals for the voltage $u_{C1}$ are achieved, but the dynamics of the current $i_{L3}$ are not good, i.e., you oscillate current in transient mode and slow decay of the ripples. An analysis of these results shows that due to the complexity of the processes in the power circuit, the application of single-criteria constrained optimization based on a standard procedure gives good results regarding the optimized parameter—in this case, the output voltage. Unfortunately, the parameter for which a limitation is introduced (current through the filter inductance) in the transient modes is observed to have large deviations from the established value of the current. This will lead to crashes, frequent tripping of the protection or large oversizing of circuit elements by current.

## 5. Second Optimization Problem

In order to limit the deviation from the established value and to improve the behavior of the circuit, a new optimization problem is formulated with respect to the current $i_{L3}$. For this purpose, a second optimization criterion has been added. The goal is to minimize the difference between the standard trajectory of the current through the filter inductance $i_{L3,ref}$ and the value of the output current obtained through numerical experiments $i_{L3}$, i.e., the following functional will be minimized:

$$J_2 = \int_0^{t_{end}} \left( i_{L3} - i_{L3_1, ref} \right)^2 dt \underset{(C_1, C_2, L_3)}{\to} \min \tag{16}$$

where the reference curve is:

$$i_{L3, ref} = 11.4(1 - e^{-t/T}), \; for \; t \in [0, 0.02] \; and \; T = 0.003 \tag{17}$$

The reference curve for the output current is chosen for the same reasons as that for the output voltage. The goal is to obtain an aperiodic transient process with a view to avoiding overloads of circuit elements in transient and dynamic modes. Thus, a multi-objective optimization problem is obtained, in which the vector of objectives is minimized:

$$J = [J_1, J_2] \underset{(C_1, C_2, L_3)}{\to} \min \tag{18}$$

Problem (18) is subject to the same constraints (12) and (15).

Since $J(x)$ is a vector, if the components of $J(x)$ are dependent, there is no unique solution to this problem. Instead, the concept of noninferior solution, also called Pareto optimality, is applied. In this case, a noninferior solution is one for which an increase in one component requires a decrease in another.

One solution to this task is through the "goal attainment method". This method was proposed by Gembicki [45]. The Goal Achievement Method is an optimization approach that focuses on achieving a set goal or set of goals. This method is used in many other fields outside of engineering, such as project management, personal development and optimal time management.

Here are some of the main steps in the method to achieve the goal:

1. Goal definition: The initial step of the method is to clearly define the goal or goals to be achieved. Goals should be specific, measurable, achievable, relevant and time-bound (SMART).
2. Goal Analysis: Examining the current situation and the resources available and assessing the obstacles that may arise on the way to the goal. Conduct a SWOT analysis (Strengths, Weaknesses, Opportunities, and Threats analysis) to assess internal and external factors related to goal achievement.
3. Strategy Development: Based on the goal analysis, develop a strategy or action plan to help achieve the goal. Breaking down the big goal into smaller, actionable sub-goals and identifying specific steps and actions to take.

4. Implementation: Implementation of the first step of the algorithm and start of implementation of the action plan. Constant monitoring and corrections if necessary. Continuation of the action and gradual approach to the goal.
5. Evaluation: Evaluation of the results of your actions and comparison with the set goal.
6. Iteration and adjustment: If the goal is not achieved, analysis of the reasons for the failure and change of strategy or approach. Repeating the optimization cycle until the desired results are achieved.

The goal attainment method provides a systematic approach to optimization by focusing on specific goals and the steps to be taken to achieve them.

It involves determining optimal design goals, $J^* = [J1^*, J2^*]$ and the relative under- or over-attainment of the goals, the latter controlled by a vector of weighting coefficients, $w = [w_1, w_2]$.

Thus, the multicriteria optimization problem is reduced to the following optimization problem:

$$\gamma \underset{(C_1,C_2,L_3)}{\rightarrow} \min \tag{19}$$

subject to:

- constrains $J(C_1, C_2, L_3) - w\gamma \leq J^*$;
- constrains (12);
- limits (15).

This optimization problem cannot be solved with a Simulink/MATLAB built-in procedure. For this, this new optimization problem is solved with MATLAB code. For this purpose, an author's program (m-file) is compiled.

In the source code of the developed program, the optimization is completed with the command fgoalattain, i.e.,

goal = [0 0]
weight = [1 1]
[x,Fval,attainfactor] = fgoalattain(@Obje,x0,goal,weight,[],[],[],[],xlb,xub)

The integrals (15) and (16) are part of the @Obje function. Furthermore, these integrals are replaced by a sum of squares of the form $\int (\ )^2 dt = \sum (\ )^2$. The differential equations (12) are also solved in this function; this is executed with solver ode23tb.

The upper and lower limits of the parameters (15) are set in the variables xlb and xub. After executing the program, the output shown in Figure 8 is obtained.

| Iter | F-count | Attainment factor | Max constraint | Line search steplength | Directional derivative | Procedure |
|---|---|---|---|---|---|---|
| 0 | 5 | 0 | 930.804 | | | |
| 1 | 14 | -3194 | 3485 | 0.125 | -1 | |
| 2 | 30 | -3194 | 3484 | 0.000977 | 0.779 | Hessian modified |
| 3 | 47 | -3194 | 3488 | 0.000488 | 0.176 | Hessian modified |
| 4 | 62 | -3194 | 3486 | 0.00195 | -1.48e-005 | Hessian modified twice |
| 5 | 71 | -3194 | 3566 | 0.125 | 0.00841 | Hessian modified twice |
| 6 | 77 | -3194 | 3397 | 1 | 0.00367 | Hessian modified |
| 7 | 83 | -3194 | 3365 | 1 | -8.69e-006 | Hessian modified |
| 8 | 90 | -3194 | 3360 | 0.5 | -2.03e-006 | Hessian modified twice |
| 9 | 146 | -3194 | 3360 | -8.88e-016 | -1.22e-005 | |
| 10 | 202 | -3194 | 3360 | -8.88e-016 | -1.22e-005 | Hessian not updated |
| 11 | 258 | -3194 | 3360 | -8.88e-016 | -1.22e-005 | Hessian not updated |
| 12 | 314 | -3194 | 3360 | -8.88e-016 | -1.22e-005 | Hessian not updated |
| 13 | 370 | -3194 | 3360 | -8.88e-016 | -1.22e-005 | Hessian not updated |
| 14 | 405 | -3194 | 3366 | 1.86e-009 | -1.22e-005 | Hessian not updated |

**Figure 8.** Results of the stepwise iteration procedures for determining optimal values of circuit elements of ZVS parallel single-ended DC-DC converter.

Optimal values for the elements are:
$C_1 = 30$ µF, $C_2 = 186.80$ µF and $L_3 = 114.75$ µH.

With the obtained optimal values, numerical experiments were performed with the model and graphical results were obtained regarding the output current and voltage of the device, which are shown in Figure 9.

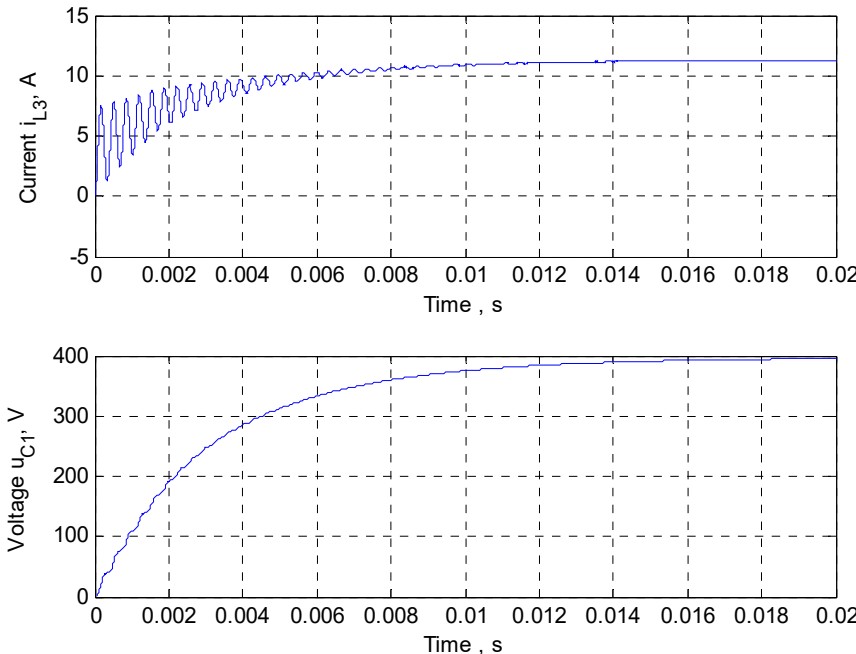

**Figure 9.** Results of simulation studies of a power elecronics converter, applying a two-criteria optimization procedure: **above**—current through the output inductance, and **below**—device output voltage.

The behavior of the ZVS parallel single-ended DC-DC converter when changing the load and the input voltage is investigated. Figure 10 shows the results of simulations with an optimized DC-DC converter at different values of load resistance *R*. Figure 11 shows the results of numerical simulations with an optimized DC-DC converter when changing the input voltage $U_d$.

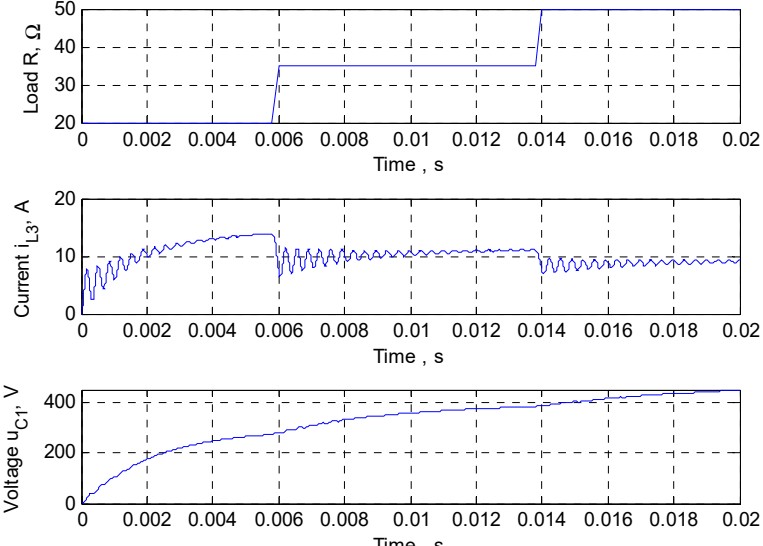

**Figure 10.** Current through the filter inductance and output voltage of a ZVS parallel DC-DC converter, at as the load resistance *R* changes.

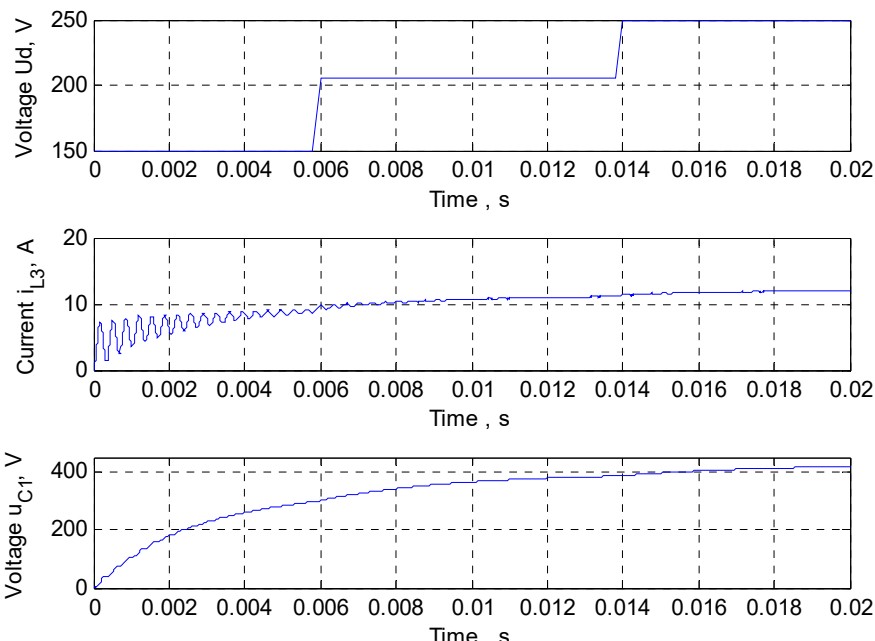

**Figure 11.** Current through the filter inductance and output voltage of a ZVS single-ended parallel DC-DC converter, at changing the input voltage $U_d$.

From the analysis of the graphical results, it is found that even with more than doubling of the load resistance value, the operation of the DC-DC converter is stable. Of course, in the transient operating modes, the current ripples increase compared to those in the established mode, but nevertheless the limitation set by the optimization regarding the output current is not violated. On the other hand, despite the very large, jumpy variation of the load relative to its nominal value, the set aperiodicity with respect to the shape of the output voltage is preserved. The above is proof of the successful execution of the optimization procedure for device design.

On the other hand, an additional advantage of the studied topology compared to the others is also established: thanks to the control method based on monitoring the maximum current through the transistor, the change of the input voltage practically does not affect the output current. This is also one of the great advantages of power circuits and the method of their control. The presented results confirm the excellent qualities of the studied power scheme.

## 6. Discussion

The analysis of the obtained results of numerical experiments shows that the use of optimization techniques achieves very good dynamic performance of the power electronic devices. Unfortunately, the use of ready-made optimization tools in Matlab/Simulink does not give good results, and therefore a specially developed author's program is used to achieve the set dynamic indicators of the device. This requires the designers to have certain basic knowledge related to the use of mathematical software and programming. In practice, thanks to the qualities of the selected power scheme, the device behaves very stably and is stable under the influence of a number of destabilizing factors such as changes in the load and the voltage of the input DC power source. This makes parallel single-ended transistor circuits very suitable for application in various vehicles, including static and dynamic charging. Typically, power circuit analyzes make a number of assumptions and design does not take into account transients and device dynamics. Analytical dependences are thus obtained, which do not take into account the type and nature of the transient processes in the power scheme. Of course, it is possible to use an optimal control synthesis and through it to obtain the quality indicators set during design. Unfortunately, both classical and modern innovative control methods applied in power electronics are influenced to a very

large extent by the qualities of the devices themselves to be controlled. In this sense, the power scheme optimally designed from the point of view of good dynamic indicators will be more stable and with the possibility of fast and reliable control synthesis. On the other hand, more sophisticated control methods based on the application of artificial intelligence techniques require the use of signal processors and the development of special software. In this way, the optimally designed power scheme will help and improve both the control synthesis and the economic and operational characteristics of the power electronic devices and systems. The proposed approach is based on achieving optimal qualities of the power circuits, thereby improving their other indicators. In studies presented in [37], it was proven that model-based optimization, by achieving certain dynamics of the device, improves in addition to its behavior during the transient process and the dependence of the operating mode on the tolerances of the resonant elements.

On the other hand, the main goal of the present work is to propose an optimization procedure that is applicable to all types of power electronic devices, regardless of their operating modes, used element base and circuit technical features. In this regard, the often-applied optimization with the objective function of achieving maximum efficiency is specific to each individual device and topology, since different analytical and graphical dependencies are used to determine the losses in the circuit elements. Another advantage of device dynamics-based optimization over other types of objective functions is that it is not always possible to precisely analytically determine element losses or other metrics used for optimization. In this way, this would lead to an incorrect optimization and, accordingly, as a final result, an optimally designed device according to the chosen criterion would not be obtained.

## 7. Conclusions

The manuscript proposes to use the combination of simplified design methodology and optimization techniques when designing circuits with complex topologies. In this way, overloading of the circuit components during transient processes is avoided. With the presented approach, it becomes possible to design topologies for which it is difficult to derive analytical relationships and/or to create and apply design procedures. In this sense, optimization is a powerful tool to support the design of power electronics converters and systems with guaranteed performance. It is important to note that ensuring the performance of power electronic devices with vehicle applications is a complex process that requires a combination of design, manufacturing, testing and after-sales service. This is completed in order to ensure high reliability and functionality of the devices, as well as to fulfill the requirements of the users and the applications for which they are intended. Model-based optimization is an important part of model-based design and its application in the design, development, implementation and prototyping of power electronic devices ensures quality reliable products at competitive prices. On the other hand, this approach allows formalization and unification of the design activity in power electronics, despite the wide variety of topologies and operating modes.

The main conclusion of the design method based on optimization techniques is that it is suitable for use in cases of research of complex topologies, modeling of electromagnetic processes with high order differential equations and transition between several stages during the control period. The presented approach is an alternative to model reduction, which is also very often applied to simplify the description of complex structures [46,47]. From the comparison of the two design approaches, it was concluded that the reduction in models gives good results and, accordingly, the reduced model is equivalent to the full one under the fulfillment of certain conditions and restrictions. In this sense, with the reduced model, we do not have complete and unconditional equivalence with the real model, and accordingly, the desired qualities and properties of the devices would not always be obtained.

The presented design approach combines traditional design techniques, developing and supplementing them with modern information and computer technologies. The

possibility of algorithmization and unification of design methodologies allows the creation of consumer software that serves to automate design. In this aspect are also the intentions for the development of research.

**Author Contributions:** N.H. and B.G. were involved in the full process of producing this paper, including conceptualization, methodology, modeling, validation, visualization and preparing the manuscript. All authors have read and agreed to the published version of the manuscript.

**Funding:** This research was funded by the Bulgarian National Scientific Fund, grant number КП-06-Н57/7/16.11.2021, and the APC was funded by КП-06-Н57/7/16.11.2021.

**Data Availability Statement:** Not applicable.

**Acknowledgments:** This research was carried out within the framework of the projects: "Artificial Intelligence-Based modeling, design, control, and operation of power electronic devices and systems", КП-06-Н57/7/16.11.2021, Bulgarian National Scientific Fund.

**Conflicts of Interest:** The authors declare no conflict of interest.

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
