# Peer review of "Design Consideration of ZVS Single-Ended Parallel Resonant DC-DC Converter, Based on Application of Optimization Techniques"

_energies, doi:10.3390/en16145295_

Round 1
Reviewer 1 Report
In this paper, the design of ZVS Single-Ended parallel resonant DC-DC Converter using optimization techniques is presented.
The procedure is promising, anyway, some parts are not clear and some improvements are required:
1. The introduction should be improved. Since in this paper, a design procedure is proposed, in the introduction some different approaches available in literature should be included describing the difference with respect to the proposed approach. It should be mentioned that analytical design approaches are usually extrapolated assuming linearities and completely neglecting the tolerance of the components. Is the proposed technique potentially suitable to take into account the non-ideality of components? What is the difference with respect to already existing methods, such as:
- Marian K. Kazimierczuk et al., "Probabilistic evaluation of power converters as support in their design", IET Power Electronics, February 2021, https://doi.org/10.1049/iet-pel.2020.0828.
- Zhou, D., Wang, H., Blaabjerg, F.: ‘Mission profile based system-levelreliability analysis of DC/DC converters for a backup power application’,IEEE Trans. Power Electron., 2018, 33, (9), pp. 8030–8039
- Owen, H., Wilson, T., Feng, S., et al.: ‘A computer-aided design procedurefor flyback step-up DC-to-DC converters’, IEEE Trans. Magn., 1972, 8, (3),pp. 289–291, doi: 10.1109/TMAG.1972.1067294
2. All the parameters in (1) to (6) should be described. Is the analytical analysis carried out neglecting all the parasitics?
3. The components on page 13 should be expressed in uH, mOhm...
4. How does the standard trajectory of the current through the filter inductance iL3,ref has been defined? It should be clarified.
5. One of the most important aspects of converters is efficiency. Why hasn't an optimization been done in that sense?
6. Since the authors propose a design procedure for a resonant converter, the achievement of ZVS and ZCS after the optimization should be shown.
Moderate editing of English language required
Author Response
First of all, I would like to thank you for your thorough review of our paper (energies-2466760) and helpful comments to improve it.
Reviewer 1
Comments to the Authors
In this paper, the design of ZVS Single-Ended parallel resonant DC-DC Converter using optimization techniques is presented.
The procedure is promising, anyway, some parts are not clear and some improvements are required:
- The introduction should be improved. Since in this paper, a design procedure is proposed, in the introduction some different approaches available in literature should be included describing the difference with respect to the proposed approach. It should be mentioned that analytical design approaches are usually extrapolated assuming linearities and completely neglecting the tolerance of the components. Is the proposed technique potentially suitable to take into account the non-ideality of components? What is the difference with respect to already existing methods, such as:
- Marian K. Kazimierczuk et al., "Probabilistic evaluation of power converters as support in their design", IET Power Electronics, February 2021, https://doi.org/10.1049/iet-pel.2020.0828.
- Zhou, D., Wang, H., Blaabjerg, F.: ‘Mission profile based system-levelreliability analysis of DC/DC converters for a backup power application’,IEEE Trans. Power Electron., 2018, 33, (9), pp. 8030–8039
- Owen, H., Wilson, T., Feng, S., et al.: ‘A computer-aided design procedurefor flyback step-up DC-to-DC converters’, IEEE Trans. Magn., 1972, 8, (3),pp. 289–291, doi: 10.1109/TMAG.1972.1067294
- All the parameters in (1) to (6) should be described. Is the analytical analysis carried out neglecting all the parasitics?
- The components on page 13 should be expressed in uH, mOhm...
- How does the standard trajectory of the current through the filter inductance iL3,ref has been defined? It should be clarified.
- One of the most important aspects of converters is efficiency. Why hasn't an optimization been done in that sense?
- Since the authors propose a design procedure for a resonant converter, the achievement of ZVS and ZCS after the optimization should be shown.
To Reviewer 1:
Thank you very much for your review and valuable remarks.
- The introduction should be improved. Since in this paper, a design procedure is proposed, in the introduction some different approaches available in literature should be included describing the difference with respect to the proposed approach. It should be mentioned that analytical design approaches are usually extrapolated assuming linearities and completely neglecting the tolerance of the components. Is the proposed technique potentially suitable to take into account the non-ideality of components? What is the difference with respect to already existing methods, such as:
- Marian K. Kazimierczuk et al., "Probabilistic evaluation of power converters as support in their design", IET Power Electronics, February 2021, https://doi.org/10.1049/iet-pel.2020.0828.
- Zhou, D., Wang, H., Blaabjerg, F.: ‘Mission profile based system-levelreliability analysis of DC/DC converters for a backup power application’,IEEE Trans. Power Electron., 2018, 33, (9), pp. 8030–8039
- Owen, H., Wilson, T., Feng, S., et al.: ‘A computer-aided design procedurefor flyback step-up DC-to-DC converters’, IEEE Trans. Magn., 1972, 8, (3),pp. 289–291, doi: 10.1109/TMAG.1972.1067294.
- Thank you very much for your comment and suggestion! The introduction has been revised and new relevant literature sources have been added. The problems with using classic design methods and the new opportunities that modeling and optimization provide are clarified.
- All the parameters in (1) to (6) should be described. Is the analytical analysis carried out neglecting all the parasitics?
- Thank you very much for the comment. Analytical dependences are derived neglecting the losses and tolerances of the circuit elements. It is for this reason that we apply the optimization to solve the problems that arise under these assumptions, and also because the analyzes were carried out in a fixed mode of operation - and thus do not account for the dynamics of the power scheme.
- The components on page 13 should be expressed in uH, mOhm...
- Thank you very much for the remark! The corresponding correction has been made.
- How does the standard trajectory of the current through the filter inductance iL3,ref has been defined? It should be clarified.
- Thank you very much for the remark! The reference curve for the output current is chosen for the same reasons as that for the output voltage. The goal is to obtain an aperiodic transient process with a view to avoiding overloads of circuit elements in transient and dynamic modes.
- One of the most important aspects of converters is efficiency. Why hasn't an optimization been done in that sense?
- Thank you very much for the remark. The aim of the present work is to propose an optimization procedure that is applicable to all types of power electronic devices. Optimization related to achieving maximum efficiency is always specific to each individual device and topology, as different analytical and graphical dependencies are used to determine losses. On the other hand, it is not always possible to accurately determine the losses analytically, and this would lead to an incorrect optimization and, accordingly, an optimally designed device would not be obtained. This was added when editing the manuscript.
- Since the authors propose a design procedure for a resonant converter, the achievement of ZVS and ZCS after the optimization should be shown.
- Thank you very much for the remark. The device operates in ZVS mode, which is guaranteed by the control system and is reflected in the device model presented in Figure 3. Therefore, no such graphs are given to demonstrate soft-switching operation.
Thank you very much for your remarks and comments. They were very useful for me to emphasize the main tasks and contributions of the manuscript, and also to focus the attention of the readers on the new and unique elements.
Reviewer 2 Report
The reviewer would like to suggest the following to improve the quality of the paper:
1- Optimization problem definition is not clear (Illustrate clearly variables, objectives and constraints). Also mention the method which MATLAB use to solve the problem.
2- The main contributions of the paper isnot mentioned properly
3- Please replace MATLAB drawings especially those in Figure 3 with more clear ones
4- A separate discussions section is required
5- Illustration is required to MATLAB lines shown to make it easy for readers who are interested in applying similar studies.
English is ok for me
Author Response
First of all, I would like to thank you for your thorough review of our paper (energies-2466760) and helpful comments to improve it.
Reviewer 2
Comments to the Authors
The reviewer would like to suggest the following to improve the quality of the paper:
1- Optimization problem definition is not clear (Illustrate clearly variables, objectives and constraints). Also mention the method which MATLAB use to solve the problem.
2- The main contributions of the paper isnot mentioned properly
3- Please replace MATLAB drawings especially those in Figure 3 with more clear ones
4- A separate discussions section is required
5- Illustration is required to MATLAB lines shown to make it easy for readers who are interested in applying similar studies.
To Reviewer 2:
Thank you for your review and valuable remarks.
- Optimization problem definition is not clear (Illustrate clearly variables, objectives and constraints). Also mention the method which MATLAB use to solve the problem.
- Thank you very much for your comment. The goal is to propose an optimization procedure that is applicable to all types of power electronic devices, regardless of their operating modes, used element base and circuit technical features. In this regard, the often applied optimization with the objective function of achieving maximum efficiency is specific to each individual device and topology, since different analytical and graphical dependencies are used to determine the losses in the circuit elements.
- The main contributions of the paper isnot mentioned properly.
- Thank you very much for your remark. An edit has been made to better present and clarify the main achievements of the manuscript.
- Please replace MATLAB drawings especially those in Figure 3 with more clear ones.
- Thank you very much. Figure quality has been improved.
- A separate discussions section is required.
- Thank you very much for your comment. A new discussion section has been added, where the obtained results are analyzed and commented.
- Illustration is required to MATLAB lines shown to make it easy for readers who are interested in applying similar studies.
- Thank you very much for your comment! My co-author Prof. Gilev and I are preparing the next manuscript, where different optimizations of power electronic devices and systems will be presented step by step with a view to achieving optimal design, using Matlab. Furthermore, the obtained results will be compared in terms of accuracy, convergence, speed of execution and hardware requirements.
Thank you very much for your remarks and comments. They were very useful for me to emphasize the main tasks and contributions of the manuscript, and also to focus the attention of the readers on the new and unique elements.
Round 2
Reviewer 1 Report
The paper has been significantly improved and it is now suitable for publication.
Reviewer 2 Report
Paper can be published in current form